# Merlin L48 Spectrogram Dataset

**Aaron Sun    Subhransu Maji    Grant Van Horn**
Manning College of Information and Computer Sciences
University of Massachusetts Amherst, MA 01003, USA
{aaronsun,smaji,gvanhorn}@umass.edu

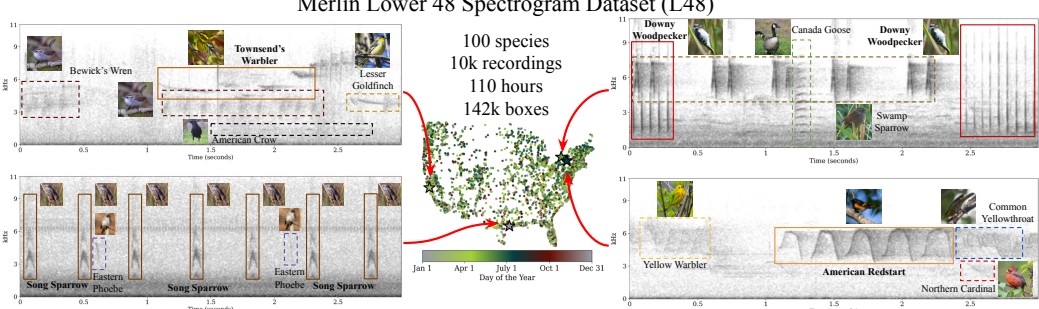

Figure 1: **The Merlin L48 Spectrogram (L48) dataset spans the Lower 48 states of the US with bird recordings throughout the year.** Each recording is associated with a target species (solid) but also contains background species (dashed), giving rise to a natural single-positive, multi-label (SPML) task. L48 stands out among similar datasets as being at country-wide, year-round scale while still maintaining high-quality bounding box annotations (see Table 1a).

## Abstract

In the single-positive multi-label (SPML) setting, each image in a dataset is labeled with the presence of a single class, while the true presence of other classes remains unknown. The challenge is to narrow the performance gap between this partially-labeled setting and fully-supervised learning, which often requires a significant annotation budget. Prior SPML methods were developed and benchmarked on synthetic datasets created by randomly sampling single positive labels from fully-annotated datasets like Pascal VOC, COCO, NUS-WIDE, and CUB200. However, this synthetic approach does not reflect real-world scenarios and fails to capture the fine-grained complexities that can lead to difficult misclassifications. In this work, we introduce the L48 dataset, a fine-grained, real-world multi-label dataset derived from recordings of bird sounds. L48 provides a natural SPML setting with single-positive annotations on a challenging, fine-grained domain, as well as two extended settings in which domain priors give access to additional negative labels. We benchmark existing SPML methods on L48 and observe significant performance differences compared to synthetic datasets and analyze method weaknesses, underscoring the need for more realistic and difficult benchmarks.

## 1   Introduction

Techniques for training multi-label models with single-positive annotations have recently gained traction in the computer vision community [2, 7, 17, 18, 46, 47]. To study this "single-positive, multi-label" (SPML) problem, researchers have adapted existing object detection datasets [11, 23]

39th Conference on Neural Information Processing Systems (NeurIPS 2025) Track on Datasets and Benchmarks.

and attribute classification datasets [5, 41] by discarding all but a single object class per sample, treating it as the single positive label. Using these modified datasets as benchmarks, researchers have proposed novel loss functions [18, 46, 47] and pseudo-labeling strategies [7, 18, 47] to handle missing labels. Impressively, top methods have closed the performance gap to fully-supervised methods in "easy" scenarios, and reduced the gap by nearly 60% in more challenging cases [18, 47].

Despite this progress, SPML methods remain underused in real-world settings such as species range map estimation (where observations typically report only one species [8]) and acoustic detection (which often includes weak labels for a single focal species [4, 33]). In practice, researchers frequently treat unknown labels as negatives rather than leveraging SPML-specific algorithms. This raises an important question: Do existing SPML benchmarks fail to capture the complexity and structure of real-world SPML scenarios? Moreover, the strict assumption of a single-positive label may be overly limiting – domain knowledge or ecological priors often allow for partial deduction of negative or even additional positive labels, but the best way of incorporating such information remains unclear.

One genuine SPML context appeared in the development of a bird sound recognition system [26]. Most human expertise in this field is region-specific, based on familiarity with a subset of species in their local context (e.g., species vocalizations within their county). As a result, when experts were asked to label all species in a recording, annotation throughput decreased dramatically because of difficulty in identifying unfamiliar species. However, when asked to identify only a single species, throughput and engagement increased significantly. This streamlined approach yielded a single-positive, multi-label dataset—a tradeoff between exhaustive annotations and practical annotation speed. Even with single-positive labels, additional negative labels can be deduced using basic ecological priors. Given this, we sought to explore: How do existing SPML methods perform on a dataset like this? What is the best way to make use of additional negative labels?

To answer these questions, we constructed the Merlin Lower-48 Spectrogram Dataset (L48) using a subset of recordings from the contiguous United States, densely annotated by experts. These annotations came in the form of spectrogram bounding boxes, creating a dataset that aligns with bird identification workflows and can be analyzed in a vision SPML context. As shown in Figure 1, Table 1, and discussed further in Section 2, the L48 is a unique dataset of bird sounds that covers a country-wide, year-round scale while containing dense species labels for each recording.

Our benchmarking revealed that several SPML methods—despite strong performance on existing benchmarks such as COCO [23]—fail to outperform a simple label smoothing baseline on L48. We attribute this gap to real-world challenges such as mismatched train-test label distributions and fine-grained label ambiguities, which are not well captured by synthetic datasets. To address these challenges, we leverage the structure of L48 to explore consistency as an additional supervisory signal. We propose a regularization scheme that improves performance across nearly all SPML methods evaluated on the dataset. By capturing practical challenges inherent in real-world scenarios, L48 serves as a valuable benchmark for deployment settings where full labeling is infeasible—such as iNatSounds [4] and BirdSet [33].

In summary, our contributions are:

1. **The L48 dataset:** A new, real-world SPML dataset reflecting practical SPML scenarios in a challenging fine-grained setting. While our focus is on SPML methods, the dataset also supports future research in detection-based species ID and semi-supervised learning.
2. **Comprehensive benchmarking:** Evaluating methods on L48 and COCO in SPML and two extended single-positive settings, revealing where commonly-examined synthetic datasets and methods fall short in realistic settings.
3. **Consistency-based regularization scheme:** A method to improve prediction consistency within a recording, enhancing the performance of all prior SPML loss functions on L48.

The dataset and code for benchmarking is publicly available at: https://github.com/cvl-umass/l48-benchmarking.

## 2 Related Work

**Bird Sound Classification Datasets.** Table 1a compares L48 to existing bird sound datasets. L48 stands out for its broad spatiotemporal coverage and dense species labels, bridging the gap between

| Dataset | D (hrs) | #S | Range | Seasonality | DL |
|---------|---------|-----|-------|-------------|-----|
| L48 | 110 | 100 | Country | Year-round | ✓ |
| iNatSounds [4] | 1,551 | 5,569 | Global | Year-round | ✗ |
| BirdSet [33] | 6,877 | 10,296 | Global | Year-round | ✗ |
| Hawaii [29] | 50 | 27 | 4 Sites | Year-round | ✓ |
| NIPS4Bplus [28] | 3 | 87 | 7 Sites | Year-round | ✓ |
| CoffeeFarms [38] | 34 | 89 | 2 Sites | Fall | ✓ |
| Amazon [13] | 21 | 132 | 7 Sites | Summer | ✓ |
| SWAMP [15] | 285 | 81 | 1 Site | Spring + Summer | ✓ |
| Western US [16] | 33 | 56 | Region | Summer | ✓ |
| Sierra [6] | 17 | 21 | Region | Summer | ✓ |
| BirdVox-14SD [9] | 300 | 14 | 1 Site | Fall | ✓ |
| BirdVox-FN [25] | 48 | 25 | 1 Site | Fall | ✓ |

(a)      (b)

Table 1: **L48 and Related Datasets.** (a): Comparison of L48 with related datasets. D denotes the total duration of training and test data, #S is the number of species, and DL indicates whether dense species labels are available. L48 is the only dataset with dense labels that spans all seasons and includes diverse regions and habitats. (b): Dataset visualizations. The top plot compares the original class distributions with those under the target-only regime. The bottom plot shows the average number of positive labels per image for each dataset after removing images with zero labeled positives.

large, archive-scale datasets and smaller, task-specific collections. BirdSet [33] and iNatSounds [4] package a subset of XenoCanto [42] and iNaturalist [14] audio, respectively, into standalone datasets. Both have global scope and over 1,000 hours of training audio but provide weak labels—each recording may contain multiple species, yet only one is labeled. On the opposite end of the spectrum, a wide range of labeled datasets exist, but have a narrower scope—focusing on specialized vocalization types [9, 25, 28, 34], specific species [9, 40], or limited geographic and temporal settings [6, 13, 15, 16, 29, 37, 38]. L48 occupies the middle ground, capturing the spatiotemporal complexities of bird identification with fine-grained labels, while remaining lightweight in size and number of classes.

**SPML Datasets.** Prior work on SPML has primarily adapted existing datasets originally designed for other computer vision tasks. The most commonly used datasets are modified versions of PASCAL VOC [11], COCO [23], NUS-WIDE [5], and CUB-200 [41]. For this work, we focus on COCO [23] due to its similarity to L48 which we discuss in Section 3.1. To create an SPML dataset version, a single object is randomly selected per image as the positive label, while all others are treated as unknown. This procedure preserves the label distributions across the train and test splits—an unrealistic condition that does not hold in the L48 dataset, where train-test label distribution shifts introduce real-world complexity. This discrepancy is apparent when comparing fully-labeled and SPML class distributions in Table 1b. Several other datasets have been used less frequently in SPML research [19, 20, 22, 30], and we discuss their quirks in Appendix A.1. In comparison to these datasets, L48 provides a fine-grained, ecologically grounded benchmark that better reflects the practical challenges of SPML.

**SPML Methods.** The baseline approach for SPML treats unknown labels as negatives [7]. Two main research directions have emerged to improve beyond this naive strategy. The first focuses on mitigating false negatives (i.e., unknown labels that are actually positive) by modifying the loss function [7, 18, 47] while the second involves pseudo-labeling strategies based on model outputs [1, 2, 7, 17, 18, 44]. Techniques like label smoothing [7], ignoring samples with high losses [18], or applying alternative losses for unknown labels [47] have shown significant performance gains. Similar improvements have been realized by pseudo-labeling strategies [1, 2, 7, 17, 18, 44], where the labels of unknown classes are iteratively estimated during training based on network predictions. In this work, we evaluate both lines of SPML methods on the L48 dataset and propose a regularization scheme based on the unique structure of the L48 to enhance their performance. While other research [24, 39, 43] has explored advanced augmentation techniques and sophisticated model backbones, we focus on core SPML strategies for simplicity and reproducibility. Finally, we treat our additional settings that provide access to confirmed negatives as a natural extension to SPML, rather than as

multi-label learning with missing labels, where labels are dropped uniformly and multiple positive labels can be present in each image [48].

# 3 The Merlin L48 Spectrogram (L48) Dataset

The L48 dataset is a curated subset of the audio collection that powers the Sound ID feature in the Merlin Bird ID app [26]. The raw audio recordings are contributed by eBird users [35], who record bird sounds in the wild with a checklist documenting the birds they saw or heard. We refer to each recording and its associated metadata as an asset. Each asset is associated with a single "target" species, though multiple bird species may be audible. We sample assets using the "target species" metadata to ensure sufficient examples per species for training and evaluation.

For L48, we selected 100 bird species from the Merlin Sound ID dataset [27], each with exactly 100 assets selected based on "target species" from the contiguous United States (the "Lower 48"). These species were selected to encompass bird sound identification challenges such as confusing species and seasonal variation while retaining a manageable number of assets and species. These assets were then annotated through a web-based interface where experts identified bird species by drawing boxes on a spectrogram with dense labels for the target and background species, as in Figure 1. To emphasize higher diversity without strenuous annotation effort, we focus on labeling *segments* of many assets (*e.g.*, Figure A2), with annotators encouraged to fully label five 6-second segments per recording. We release all segments in L48 to support semi-supervised SPML research, though for this study, we train only on segments containing at least one positive label. The dataset is split into 80 assets per species for training and 20 for testing, with 10 training assets per species held out for validation.

The Merlin Sound ID models [27] use computer vision backbones that process spectrograms in real time, mirroring how human experts interpret bird sounds visually for identification. To make L48 accessible to the computer vision community, we release it as a collection of images using the same spectrogram settings applied during annotation (see Appendix A.2.1 for details). These images can have overlapping vocalizations which distort the resulting spectrogram as in Figure 2, unlike in conventional image datasets where objects entirely occlude one another. Concretely, let $\mathcal{D} = \{(\mathcal{X}_i, \mathcal{Y}_i, t_i, \mathcal{M}_i)\}_{i=1}^N$ denote the L48 dataset, where $\mathcal{X}_i$ denotes the $i$-th asset with annotations $\mathcal{Y}_i = \{b_1^i, b_2^i, \ldots, b_k^i\}$, target class $t_i \in \{1, \ldots, M\}$, and metadata $\mathcal{M}_i$ (see Appendix A.2.2 for details). Each $b_j^i = (\text{box}_j^i, \text{cls}_j^i)$ indicates a bounding box ($\text{box}_j^i$) with label $\text{cls}_j^i \in \{1, \ldots, M\}$. To convert this into a spectrogram multi-label dataset, we divide each asset into non-overlapping 3s clips, meaning each labeled asset $(\mathcal{X}_i, \mathcal{Y}_i, t_i, \mathcal{M}_i)$ is mapped to $(\mathbf{x}_k^i, y_k^i, t_i)_{k=1}^{N_i}$, where $\mathbf{x}_k^i$ denotes the $k$-th 3s clip within asset $\mathcal{X}_i$, and $y_k^i \in \{0, 1\}^M$ denotes the presence or absence of each class label within the clip. A class is marked present if the bounding box is overlapping and not mostly truncated, which we define as being longer than 80ms but only occupying the first or last 200ms. Annotations for species outside the 100 selected species are ignored and excluded from L48.

## 3.1 SPML Benchmarks and Data Regimes

**L48.** The construction of the L48 dataset naturally gives rise to three data regimes: target-only, target-only with geographical priors, and target-only with checklist priors. Each regime includes a single positive label per example, with an increasing number of negative labels. We generate these datasets for training and evaluate all models on a fixed, fully annotated test set.

In the target-only regime, we retain clips where the class corresponding to the asset's target label is present. The dataset is defined as a set of clips $\mathcal{T}_k$ for each class $k \in 1, \ldots, M$ that contain the target label, i.e., $\mathcal{T}_k = \{\mathbf{x}_j^i | t_i = k \wedge y_j^i[k] = 1\}$. For each clip, the presence of all other classes is treated as unknown. Figure 1 illustrates this setup: clips containing the target class (shown as solid boxes) are retained, while all other boxes are treated as unknown.

Additional negative labels are introduced using domain-specific priors derived from metadata. These priors provide only negative labels; presence on a checklist or within a geographic range does not imply vocalization in a specific asset.

First, geographical (geo) priors leverage the known range of each species. For example, if a recording was made in California, species like the Eastern Towhee—which do not occur in that region—can be

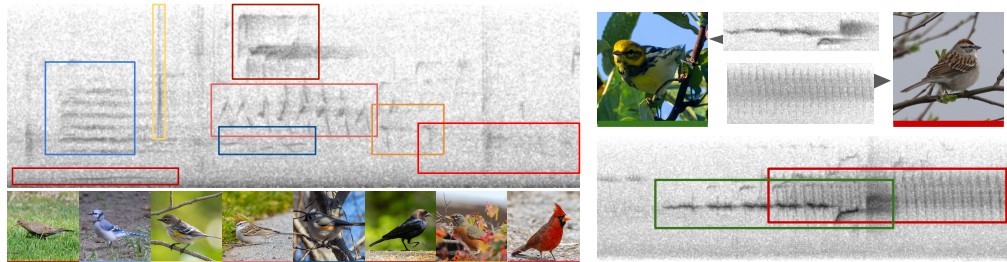

Figure 2: **An illustration of how time and frequency overlaps can cause distortions in the resulting spectrogram.** Images are underlined with corresponding box colors. Left, an image with 8 different species vocalizing (from left to right: Mourning Dove, Blue Jay, Yellow-rumped Warbler, Chipping Sparrow, Tufted Titmouse, Brown-headed Cowbird, American Robin, Northern Cardinal). Right, the vocalizations of Black-throated Green Warbler (green) and Chipping Sparrow (red) are depicted and the two birds are shown vocalizing simultaneously.

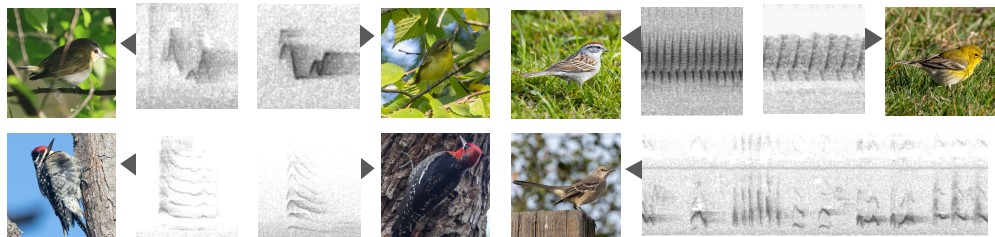

Figure 3: **Examples of difficult vocalizations in the L48.** The bottom right shows a Northern Mockingbird imitating other birds in its long song, while the others show confusing species pairs. From left to right, the top row shows Red-eyed Vireo, Philadelphia Vireo, Chipping Sparrow, and Pine Warbler songs, while the bottom row shows Yellow-bellied Sapsucker and Red-breasted Sapsucker.

excluded as negatives. Concretely, given a set of possible species $\tilde{y}_j^i \in \{0, 1\}^M$ from metadata $\mathcal{M}_i$ by range, we maintain the target species label $y_j^i[k] = 1$ and specify $y_j^i[s] = 0$ where $\tilde{y}_j^i = 0$. This approach yields an average of 42 negative labels per image across the dataset.

An even stronger prior comes from eBird [35] checklists, which accompany each asset and contain a complete list of species seen or heard at the time of recording. Because any vocalizing species must be present on the checklist, species absent from it can be confidently labeled as negatives. This checklist-based prior provides 79 negative labels per image on average, meaning images only have 20 unknown labels per image on average compared to 99 in the target-only regime.

**COCO.** For comparison, we synthetically generate target-only, geo, and checklist-style data regimes on the widely used MS-COCO 2014 (COCO) [23] dataset. First, we convert COCO from an object detection task to a multi-label classification task by discarding bounding boxes and retaining only class presence labels. To create the target-only regime, we follow the procedure from [7]: for each image, we randomly sample a single positive label and discard all others. We repeat this process using five different random seeds to produce five SPML variants of COCO for benchmarking.

To simulate geographical and checklist priors on COCO, we introduce negative labels using scene-based priors. The goal is to mimic the inference of improbable classes based on scene context—e.g., elephants are unlikely to appear in office scenes. To model class "ranges," we first assign each image a scene category vector by computing CLIP [32] similarity with the categories in the Scene-15 dataset [31]. We then train a linear model on 10% of the training data to predict full class labels from these scene similarities, avoiding overfitting. Using the trained model, we generate class predictions across the full training set. The lowest-scoring predictions are treated as negative labels, while any incorrectly marked negatives are reverted to unknown. By adjusting the score threshold to match the average number of negatives in the geographical and checklist regimes of L48 dataset, we produce datasets where 45% and 83% of class labels per image are known negatives, respectively.

**Dataset Comparison.** Table 1b gives an overview of how L48 dataset statistics compare to COCO. The training set sizes of L48 and COCO are 82,081 and 45,178, respectively, and COCO has 80 classes while the L48 has 100. These statistics reflect images which contain at least one positive label, as L48 also contains empty and unlabeled images which can be used for pre-training and other semi-supervised tasks (*e.g.*, Figure A2), which makes up a set of 40,015 additional images which we exclude from training. Though COCO typically has more positives per image than L48, this does not reflect the difficulty of each dataset. L48 is fine-grained in nature, meaning there are easily confusable species pairs which are harder to distinguish than the common classes in COCO, some of which are shown in Figure 3. Lastly, we note that the L48 can be framed as an object-detection task on spectrograms. In this work, we focus on a multi-label setting. We include additional information on comparisons specific to object detection in Appendix A.3.1.

## 4 Methods

We conduct an empirical comparison of L48 and COCO across fully supervised, target-only, and geo/checklist prior settings. Training protocol details are provided in Section 4.4 and Appendix A.4.

### 4.1 SPML Methods

In a standard multi-label task, our goal is to learn a classifier $\hat{y} = f_\theta(\mathbf{x})$, where $\mathbf{x}$ is an image, $\hat{y} \in [0,1]^M$ denotes the score for each class, and $\theta$ are the learnable parameters of the model. In this setting, typically losses are defined independently on the positive and negative labels and then averaged. Binary cross-entropy (BCE) loss defines loss for class $i$ on the positive labels $\mathcal{L}_{\text{BCE}}^+(\hat{y}[i]) = -\log(\hat{y}[i])$ and on the negative labels as $\mathcal{L}_{\text{BCE}}^-(\hat{y}[i]) = -\log(1-\hat{y}[i])$. The total loss can be averaged with labels $y \in \{0,1\}^M$ as $\mathcal{L}_{\text{BCE}}(\hat{y}, y) = \frac{1}{M}\sum_{i=1}^M y[i]\mathcal{L}_{\text{BCE}}^+(\hat{y}[i]) + (1-y[i])\mathcal{L}_{\text{BCE}}^-(\hat{y}[i])$. However, in the SPML setting, only positive and unknown labels are provided, and these methods must define losses for these two regimes instead, $\mathcal{L}_{\text{SPML}}^+$ and $\mathcal{L}_{\text{SPML}}^?$, respectively, and are averaged across labels to make $\mathcal{L}_{\text{SPML}}$.

We compare performance on the following SPML methods: 1) BCE-AN (binary cross-entropy, assume negative) [7] trains with standard BCE loss by treating all unknown labels as negative. 2) WAN (weak assume negative) [7] uses BCE-AN while downweighting the loss for all unknown labels. 3) LS (label smoothing) [36] uniformly smooths labels to reduce the impact of false negatives. 4) ROLE (regularized online label estimation) [7] creates a table of pseudo-labels which are jointly trained alongside the model. 5) EM (entropy maximization) [47] has a unique loss landscape which only encourages the model to increase confidence on already confident predictions, while ignoring uncertain ones. 6) LL (large-loss) [18] variants change the behavior of BCE-AN by reducing the penalty for confident predictions on unknown labels, which the authors speculate could be positives incorrectly assumed to be absent. LL-R (LL-rejection) sets these terms to zero, LL-Ct (LL-temporary correction) temporarily treats the label as positive, and LL-Cp (LL-permanent correction) permanently modifies the label to positive. Explicit loss definitions are provided in Appendix Table A10.

### 4.2 Asset Regularization

Our experimentation revealed existing SPML methods suffer from misclassifications, which can be mitigated by utilizing the structure of L48—each image is a cropped view from a much longer spectrogram. Similar to video action recognition [45], we posit the model should have temporal consistency throughout an asset. We propose utilizing this structure in a regularization term that enforces prediction consistency across all images from the same asset: $\mathcal{R}_P(\mathbf{x}_j^i) = \mathcal{L}_{\text{BCE}}(f_\theta(\mathbf{x}_j^i), \overline{y}_t^i)$, where $f_\theta$ is the current model, $\mathbf{x}_j^i$ is the $j$-th clip of asset $i$ and $\overline{y}_t^i$ is the average of predictions across an asset at the $t$-th training step. We update $\overline{y}_t^i$ using a simple moving average with a hyperparameter $\epsilon$: $\overline{y}_{t+1}^i = (1-\epsilon)\overline{y}_t^i + \epsilon f_\theta(\mathbf{x}_j^i)$. Based on our observation of target species recurrence, we hypothesize background species which occur within an asset similarly tend to repeat across multiple images within that asset. Qualitatively, we found species which appear at least once tend to occur in 28% of clips from that same asset. Hence, in general we expect that ground truth labels should be similar between images within the same asset. This is similar to the consistency loss introduced in [39], but we exploit the image-asset relationship present in the L48 to promote inter-image similarity rather

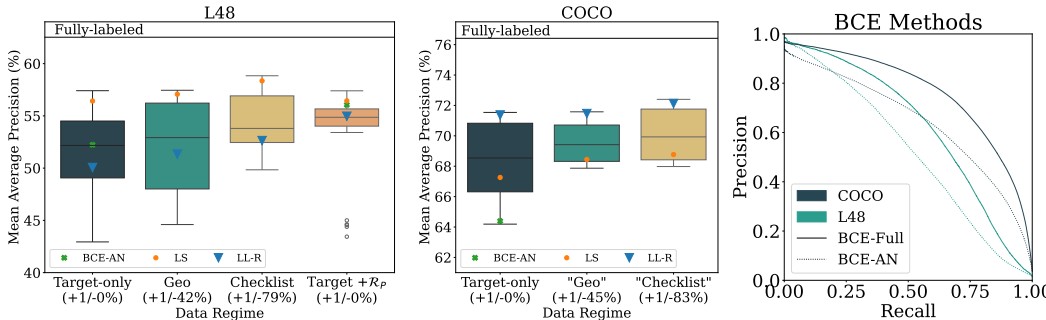

Figure 4: **Overview of results.** Left: L48 (leftmost) and COCO (middle) mAP performance distributions across five trials of each SPML method for four different data regimes, shown as box plots and lines. For each box, the thin lines shows the 1.5x interquartile range, the box shows the interquartile range, and the horizontal line shows the median. In parentheses, the proportion or number of annotated labels is given, with + signifying positive labels and - signifying negative labels. The mean performance of three methods are plotted: BCE-AN, LS, and LL-R. For L48 we show the target-only performance with asset regularization. Right: L48 (left) and COCO (right) class-averaged precision-recall curves for BCE-Full and BCE-AN.

than requiring multiple augmentations of a single image to boost performance. We combine this regularization with existing SPML losses as $\mathcal{L}_{\text{SPML}} + \alpha\mathcal{R}_P$, where $\alpha$ is a hyperparameter.

### 4.3 Incorporating Negative Label Priors

In the geographical and checklist regimes, we modify each SPML method for explicit negative labels by either adding to or entirely replacing the existing loss with $\mathcal{L}_{\text{BCE}}^-$. More explicitly, we define $\mathcal{L}_{\text{SPML}}^- = a\mathcal{L}_{\text{SPML}}^? + b\mathcal{L}_{\text{BCE}}^-$, where $a$ is either 0 or 1 and $b$ is a hyperparameter. We make an exception for the LL-variants, which can be straightforwardly modified to account for negative labels, by excluding known labels from rejection and correction. We tune $a, b$ on our validation set and take the best performing settings on each method and dataset for testing.

### 4.4 Network Architecture, Training, Hyperparameter Search

Following the training procedure of prior work [7, 18, 47], each of our experiments was trained using an ImageNet [10] pretrained ResNet50 [12] architecture. Prior works [4, 37] have shown substantial improvements from ImageNet pretraining for spectrogram classification despite the domain shift. We preprocess each image by resizing the image to shape (448, 448) and normalizing the image to ImageNet statistics. For COCO only, we flip the image horizontally at random for training. For each method we select learning rate and other hyperparameters related to the loss using the validation set and report performance on the test set. Our experiments were run for 10 epochs on NVIDIA GTX 1080 Ti, GTX 2080 Ti, and GTX Titan X with trials taking approximately 3 hours for L48 and 5 hours for COCO. Further details are in Appendix A.4.

## 5 Results

### 5.1 Fully-Supervised Performance

**The L48 is a challenging multi-label dataset.** In the first row of Table 2, we see performance on L48 is 14 points lower than on COCO despite having a similar number of images and classes. We attribute this deficit primarily to higher classification difficulty for the L48. Unlike the common objects in COCO, L48 has many fine-grained species pairs, like those in Figure 3, which leads to lower overall performance due to model misclassifications. This is most evident in Figure 4, as BCE-Full precision starts higher but falls off much more quickly for L48 than COCO.

| Method | COCO | COCO + Geo | COCO + CL | L48 | L48 + Geo | L48 + CL | L48 $+\mathcal{R}_P$ |
|---|---|---|---|---|---|---|---|
| BCE-Full | $76.4 \pm 0.1$ | — | — | $62.4 \pm 0.5$ | — | — | $66.4 \pm 0.5$ |
| BCE-AN | $64.4 \pm 0.1$ | — | — | $52.2 \pm 0.5$ | — | — | $56.1 \pm 1.1$ |
| WAN | $66.1 \pm 0.2$ | $68.4 \pm 0.2$ | $68.4 \pm 0.2$ | $52.0 \pm 0.6$ | $52.4 \pm 0.5$ | $54.0 \pm 0.3$ | $55.7 \pm 0.7$ |
| LS | $67.3 \pm 0.2$ | $68.4 \pm 0.2$ | $68.8 \pm 0.1$ | $\mathbf{56.4 \pm 0.7}$ | $\mathbf{57.1 \pm 0.3}$ | $\mathbf{58.4 \pm 0.3}$ | $\mathbf{56.4 \pm 0.7}$ |
| ROLE | $66.6 \pm 0.3$ | $68.0 \pm 0.1$ | $68.2 \pm 0.1$ | $54.0 \pm 1.0$ | $54.8 \pm 0.7$ | $54.0 \pm 0.7$ | $54.1 \pm 0.5$ |
| EM | $71.1 \pm 0.2$ | $\mathbf{71.8 \pm 0.1}$ | $\mathbf{72.3 \pm 0.2}$ | $55.3 \pm 1.0$ | $56.3 \pm 0.6$ | $57.2 \pm 0.4$ | $55.2 \pm 0.7$ |
| LL-R | $\mathbf{71.4 \pm 0.1}$ | $71.5 \pm 0.1$ | $72.1 \pm 0.2$ | $50.1 \pm 0.8$ | $51.3 \pm 0.5$ | $52.6 \pm 0.4$ | $55.0 \pm 0.4$ |
| LL-Ct | $70.5 \pm 0.2$ | $70.7 \pm 0.2$ | $71.2 \pm 0.2$ | $48.0 \pm 0.9$ | $48.1 \pm 0.9$ | $52.4 \pm 1.0$ | $54.1 \pm 0.6$ |
| LL-Cp | $69.8 \pm 0.2$ | $70.2 \pm 0.1$ | $71.8 \pm 0.1$ | $43.8 \pm 0.8$ | $45.8 \pm 1.1$ | $50.6 \pm 0.4$ | $44.4 \pm 0.6$ |
| SPML Avg | 68.4 | 69.9 | 70.3 | 51.5 | 52.2 | 54.0 | 53.9 |

Table 2: **Main results.** Compiled mAP results (given in percentages) on the test set for each method, averaged across five runs with standard deviation given. The best SPML method is given in bold and average performance is shown separately on the bottom row.

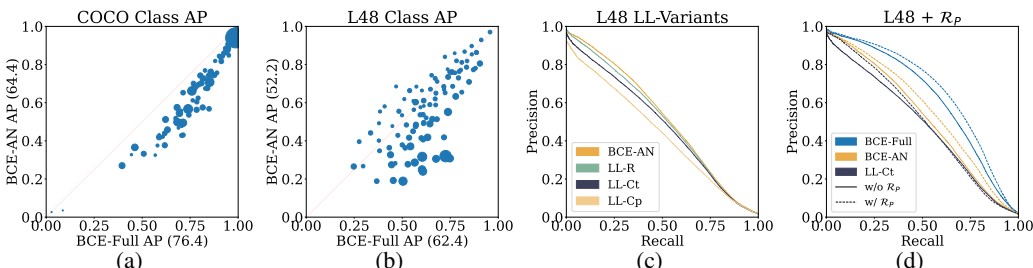

Figure 5: **In-depth method performance analysis.** (a-b): Per-class average precision on BCE-AN compared to BCE-Full for both datasets, where dot size is proportional to class frequency in the test set. Classes which perform worse in SPML training fall below the diagonal line. Method names are given in axes, with mAP in parentheses as percentages. (c-d): Precision-recall curves for various methods on various data regimes of the L48.

## 5.2 SPML Performance

**SPML methods are less effective on the L48 in part due to the target-only sampling procedure.** In Table 1b, we see target-only sampling leaves a uniform distribution for the L48 while the SPML version of COCO has a similar distribution to the original dataset. This distribution mismatch has been shown to lower performance in the corresponding SPML task [3], and our results corroborate this finding. In Figure 5, (b) shows the performance of common classes (shown as larger dots) suffers the steepest decline in average precision on L48, while on the COCO dataset in (a) the correlation between performance drop and class frequency is less apparent.

**Unlike COCO, L48 highlights incorrect assumptions made by some SPML methods when dealing with misclassifications on fine-grained classes.** Existing SPML methods aim to reduce the impact of mislabeled negatives through loss function adjustments [7, 18] or pseudo-labeling [18, 47]. However, mislabeled negatives and model misclassifications are indistinguishable when observing model outputs for a single image. This ambiguity is not apparent in COCO, where objects are rarely confused, and high model confidence on an unknown label is likely a mislabeled negative. However, the rate of misclassifications is much higher for the fine-grained classes in L48 (*e.g.*, Figure 3), and methods which interpret class confusion as an incorrect label suffer. This is exemplified for LL-variants, which assume confident predictions imply unknown labels are positive. Despite strong performance on COCO, we see in Table 2 these methods all falter on L48, dropping below BCE-AN performance. Figure 5c shows that the stronger assumptions made by LL-Ct and LL-Cp lead to more false positives and lower precision than LL-R, which ignores the loss on unknown labels instead of correcting them. In contrast, methods which do not make this assumption (EM [47] and LS [7]) perform relatively well on the L48 and stand out as the best performing methods.

### 5.3 Asset Regularization Results

**Asset regularization is beneficial for nearly all methods.** Given the lack of labels in the target-only dataset, asset regularization provides additional supervision by enforcing consistency between clips within the same recording. With the addition of this regularization term, we see in the rightmost column of Table 2 that most methods approach the performance of LS, the best-performing method without this regularization. Interestingly, in Figure 5d we see regularization improves performance even for BCE-Full, mainly for lower precisions.

**Asset regularization reduces misclassification rates by averaging many observations of the same species.** Once again, this is most apparent in large-loss methods, which outright encourage false positives by ignoring losses for these predictions. In Figure 5, we see the recall at high precisions for LL-Ct after applying asset regularization increases dramatically to a level matching BCE-AN. We hypothesize these errant misclassifications can be separated from mislabeled negatives by utilizing multiple "views" across time. Since a background species is likely to occur multiple times, misclassifications are subdued when the model can correct itself over many observations of the same species. This does not apply to LL-Cp, which permanently modifies labels on confident predictions which cannot be corrected, and performance remains poor even with regularization. These benefits are also muted on EM and ROLE, which have significantly different output distributions shown in Figure A4 and likely require a regularization tailored to handle their unique loss landscapes.

### 5.4 Geographical and Checklist Prior Results

In Table 2, we present method performance on L48 and COCO with additional negative labels generated through geographical and checklist priors described in Section 3.1.

**Average performance consistently improves with negative labels.** In Figure 4 we observe average method performance increasing across both datasets from the target-only regime to the geographical and checklist regimes. This is sensible, as additional labels should improve training in most circumstances. Interestingly, for COCO the boost from target-only to geo is larger than from geo to checklist, while for L48 the trend is reversed. This may signify a complex relationship dependent on which negative labels are revealed for a given image. Additionally, we see the same average performance in the checklist regime as in target-only with asset regularization, indicating the value of weak background species supervision nearly matches the elimination of nearly 80% of unknown labels. Taken together, these points seem to indicate strong promise for active learning in these settings—careful, directed supervision can be as valuable as broad-scale labels across the dataset.

**We see room for improvement on L48 to close the gap between target-only and BCE-Full performance.** As shown in Figure 4, the average performance gains of 1-2 points fall well short of the 8-10 points needed to match BCE-Full. Although we adapted the SPML methods in Section 4.3 for the extended SPML setting, only EM and LL-variants show consistent improvements across both datasets as negative labels are introduced. This highlights an opportunity to develop methods that are more flexible in leveraging additional labels informed by the problem context. Moreover, using negative labels is just one avenue for progress on L48—we leave the exploration of rich signals from unlabeled data and bounding box annotations for future work.

## 6  Conclusion

In this work, we thoroughly examine how the L48 dataset highlights where existing evaluations for SPML methods can overlook challenges in real-world deployment. Unlike synthetic versions of COCO, L48 has a natural SPML setting from target-only recordings and introduces additional fine-grained confusions not present in other datasets. L48 also has a unique asset-clip relationship which we utilize in asset regularization to boost performance across nearly all methods. This methodology might be extended to other settings with multiple related images per input, such as large satellite images or videos. Furthermore, the ecological setting of L48 allows for explicit negatives in two data regimes, but we find most SPML methods are not readily modified to utilize additional data. We see room for improvement in this field and leave open questions to how performance on L48 can benefit from additional data such as bounding box annotations and unlabeled data for semi-supervised learning. Our dataset also reveals the value of labeling background species for building strong

recognition models and suggests avenues for research in active learning to best leverage the targeted expertise of annotators.

While L48 comprehensively covers 100 species across the lower 48, this is a relatively limited geographical range and set of species compared to the biodiversity present throughout the world. As a result, conclusions drawn from L48 might not transfer to other regions and species sets. However, L48 is a strong benchmark which captures the complexities of real-world deployment with dense labels on a highly specialized task. Through further experimentation, we hope to demystify problems like weak labels and geographical priors for datasets such as iNatSounds [4] and BirdSet [33], which encompass species across the world. These datasets, when used responsibly in tandem, might be the keys to supporting conservation for sensitive species and threatened habitats globally.

## Acknowledgments

Subhransu Maji and Aaron Sun were supported in part by NSF grant #2329927.

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

# A  Appendix

## A.1  Other Datasets

Below we describe a few other prevalent multi-label datasets and explain how the L48 differs from them, hence they were excluded from comparison in this paper.

**PASCAL VOC** [11] was created for object detection and classification, covering 20 basic-level classes across 4,574 images, with most images containing a single prominent object. This dataset is much smaller than L48 and also contains much fewer classes which are all coarse-grained.

**VG500** is a modification of the Visual Genome dataset [19], a dataset focused on dense annotations linking images to respective captions. This dataset is not intended to be bounded by categories but has open-vocabulary annotations. To turn this into a multi-label task, only the top 500 most frequent categories are kept to make VG500, following the work in [21]. We choose not to compare to this dataset because the open-vocabulary nature of the task leaves ambiguity in annotations but no clarification is given between explicit negatives and unknowns.

**OpenImages** [20] is a large-scale dataset with 14.6M boxes across 1.7M images spanning over 600 categories. Similar to Visual Genome, a semantic hierarchy is given and both positives and negatives are given explicitly. This dataset is similar to L48 in nature, but differs entirely in scale, containing about 10 times in the number of images in the training set. Since this dataset is used in a completely different context to L48 due to the size, we chose not to compare to this dataset.

**NUS-WIDE** [5] is another multi-label dataset based on publicly-available internet images. This dataset contains images from Flickr which are labeled with corresponding tags for 81 concepts. This dataset is no longer available in its entirety due to many of the associated images being no longer accessible on Flickr. In addition, not all of the concepts are object-centric and can be associated with a bounding box, including abstract concepts such as "protest" and less clearly explicit events such as "earthquake." Based on these issues and differences from L48, we excluded NUS-WIDE in our comparison.

**WIDER-Attributes** [22] is a dataset focused on classifying human attributes, but only focuses on 14 attributes per person in an image. This task is much less fine-grained than the L48 and contains far fewer classes than L48, which led to its exclusion in our analysis.

**Caltech-UCSD-Birds** (CUB200) [41] is conventionally used as a classification dataset, but can also be treated as an attribute prediction task for each bird. However, these attributes are non-binary (such as the shape of the bill being curved, hooked, cone, etc.), so to transform this into a multi-label problem, each of these attributes must be turned into a set of binary attributes equal to the number of choices where they are mutually exclusive. This is not an object-centric task like the L48, and we believe turning multiple classification problems into a single multi-label problem is contrived so we exclude it from our comparisons.

**Visual Privacy** (VISPR) [30] is a dataset which identifies personally revealing information within images, where each category signifies whether a given personal characteristic can be found within an image. While some of these attributes are explicit to identify such as phone number and eye color, others are abstract, such as religion, personal relationships, and hobbies. We primarily exclude this from our analysis because the labels are not object-centric like in L48 and are more difficult to interpret.

## A.2  L48 Additional Information

We organize the L48 by images in sets which come from recordings, which we also call clips and assets, respectively. We outline the metadata associated with each image and recording as well as our spectrogram generation process below.

### A.2.1  Spectrogram Generation

To generate spectrograms from 1D waveforms, we use the Short-Time Fourier Transform with a window size of 512 and stride length of 128. This spectrogram is then converted to individual images which span 3 seconds and are disjoint. To input the spectrogram into our network, we copy the spectrogram into three channels and resize it to shape $448 \times 448 \times 3$.

| Field | Possible Values | Description |
|---|---|---|
| id | [0, 9999] | The unique ID associated with the asset |
| split | [train, test] | Denotes training split or test split for an asset |
| target_species_code | 6-letter-code | The target species for this asset |
| possible_species_codes | [6-letter-codes] | A list of possible species based on ranges |
| observed_species_codes | [6-letter-codes] | A list of species in the affiliated checklist |
| present_species_codes | [6-letter-codes] | A list of positively labeled species |
| unknown_species_codes | [6-letter-codes] | All species not in present or absent lists |
| absent_species_codes | [6-letter-codes] | A list of negatively labeled species |

Table A1: A summary of asset metadata and their possible values.

| Field | Possible Values | Description |
|---|---|---|
| id | [0, 416534] | The unique ID associated with the clip |
| asset_id | [0, 9999] | The asset ID from which this clip came |
| clip_order | [0, 1449] | The position of the clip within the asset |
| file_path | Relative filepath | The path to the image for the given clip |
| width | 750 | The image width |
| height | 236 | The image height |
| present_species_codes | [6-letter-codes] | A list species with positive labels |
| unknown_species_codes | [6-letter-codes] | All species not in present or absent lists |
| absent_species_codes | [6-letter-codes] | A list species with negative labels |
| boxes | [dictionaries] | Bounding box annotations for the clip, see Table A3 |

Table A2: A summary of clip metadata and their possible values.

### A.2.2 Asset Metadata

Assets have associated metadata which we summarize in Table A1 and also explain in detail below.

Each asset is associated with a unique asset ID from 0 to 9999. Assets with an ID greater than or equal to 8000 are test assets, and each species has 80 training assets and 20 test assets. For our experiments, we randomly selected 10 training assets per species to serve as validation assets for hyperparameter tuning (given in the repository). Each asset contains a variable number of clips, with a minimum of 11 and a maximum of 1450. As discussed in the paper, every asset has a target species which is provided in the form of a 6-character target species code. The corresponding taxonomic information such as phylogeny, common name, and scientific name are given in taxa.csv.

Assets also contain compiled lists of positives, negatives, and unknowns, where positives are also known as present species and negatives are also known as absent species. The list of positives is the union of positives given across each clip in the asset, while the list of negatives is the intersection of clip negatives. The list of unknown species contains the species which are not in either of the previous two lists.

Assets also contain two additional fields, possible species given by geographic priors and observed species within the associated checklist. Using the location and time of year each recording was taken, we are able to generate a list of possible species based on species ranges. Though this list does not provide positive labels, absence of a species on this list implies a negative label for that species across the entire recording. This logic also applies for observed species within the associated checklist. Any species present in the recording should also be reported in the associated checklist, so species not on the checklist should have negative labels for the recording. The negative labels generated through checklist data is a superset of the negative labels generated from geographical priors. Hence, geographical priors and checklist data provide two additional levels of weak supervision which falls between SPML and full-labels. We apply negative labels from geographical and range priors to the clip level, even for unlabeled data.

| Field | Possible Values | Description |
|---|---|---|
| `id` | int | Box ID unique to each clip |
| `species_code` | 6-letter-code | The species which this vocalization belongs to |
| `status` | ["passive", "active", "ignore"] | Species prevalence in the clip |
| `bbox` | $[0, 1]^4$ | Box coordinates [xmin, ymin, xmax, ymax] |

Table A3: A summary of box data and their possible values.

| Dataset | Boxes/image | Small | Medium | Large |
|---|---|---|---|---|
| VOC | 3.28 | 2.96% | 19.79% | 77.24% |
| COCO | 9.17 | 19.95% | 34.36% | 45.69% |
| L48 + | 2.38 | 0.97% | 7.85% | 91.18% |

Table A4: An overview of each datasets' box statistics in terms of sizes and quantities for the training set. To standardize which boxes are small, medium, and large, we resize each image and its bounding boxes such that the minimum dimension of the image is 640, then we threshold by bounding box area. Small boxes have area less than $32^2$, large boxes have area greater than $96^2$, and all other boxes are medium boxes. L48 + signals images with no boxes are not considered.

### A.2.3 Clip Metadata

Clips also have corresponding metadata which is summarized in Table A2. The bounding box annotations for each clip are provided, where each box is specified with an ID, species code, status, and coordinates. The box ID is unique to a clip, so no two boxes within the same clip share the same ID. The bounding box coordinates are given in relative coordinates falling within [0, 1] and are provided as [xmin, ymin, xmax, ymax]. For box status, sounds which are longer than 80 ms which are only present in the first or last 200 ms of a window are labeled "ignore" while others are "active."

Boxes which do not have status "ignore" are treated as positive labels for the multi-label task and are given in the list of positives. Any clip with positive labels is treated as fully-labeled, meaning all other species are negative, unless there are "Unknown bird" boxes, in which case we put treat other possible species as unknown (but retain negatives from geographical priors).

### A.3 Additional Dataset Statistics

In this section, we compare additional statistics of the L48 to VOC and COCO not covered in the main paper.

### A.3.1 Bounding Box Statistics

We give basic statistics of bounding boxes quantity and sizes in Table A4. L48 is most similar to VOC among the datasets which we compare to. On average, an image contains 2.25 boxes, and the vast majority of these boxes are usually large. This likely occurs because most vocalizations in a spectrogram span a wide range of frequencies due to overtones, so most boxes have a height comparable to the image height. The duration of these vocalizations can vary, depending on whether they encompass a single call or a longer bird song. These distributions are also visualized in Figure A1.

### A.3.2 Known and Unknown Label Statistics

We give statistics for the breakdown of images which are fully-labeled, images which contain at least one positive label, and images with any labels in Table A5. Though negative labels are generated for all images using the metadata outlined in Section A.2.2, bounding boxes are all hand-drawn by expert annotators, who focus on annotating various segments of a recording instead of the entire thing. Hence, only 45% of the training set is fully-annotated for all species. Our original data contained "Unknown Bird" boxes for vocalizations which were unable to be identified to species level. As a result, we cannot generate negative labels reliably for these images, and they remain partially-labeled despite containing positive labels. We train our model with all images with at least one positive

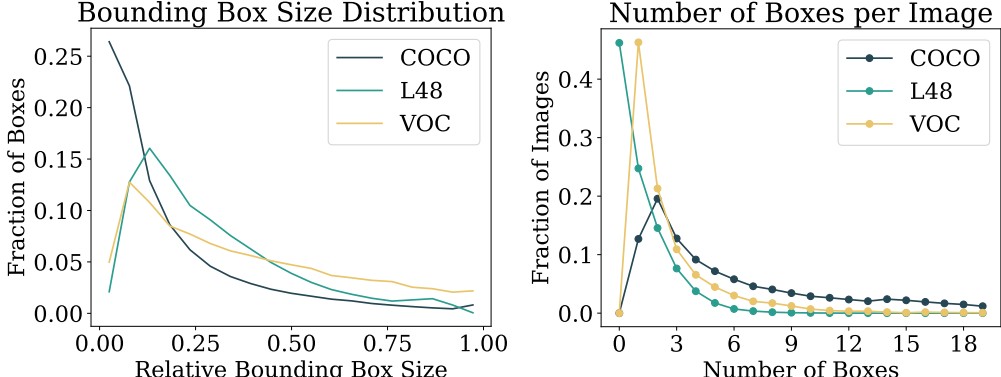

Figure A1: Visualization of bounding box distributions for each dataset. The left plot shows the bounding box size distribution, where the relative size gives the area of the box divided by the total image area. The right box shows the number of boxes per image. L48 mirrors the distribution of VOC closely in terms of boxes per image, but has a unique bounding box size distribution.

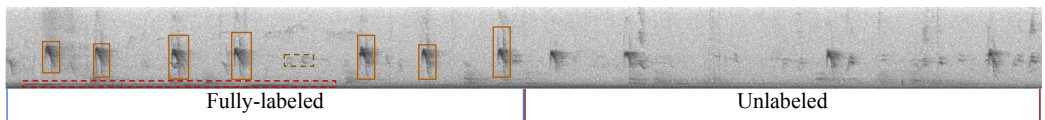

Figure A2: An excerpt from a partially-labeled asset in L48. The first half of this snippet is fully-labeled while the last half is unlabeled. For our experiments we train only on the first half, but we release the full asset for future work on semi-supervised and unsupervised learning. The vocalizing birds are Mourning Dove, Canyon Wren, and House Finch in order of first appearance from left to right.

label, which is 53% of the dataset. We do not use the remaining data for training in this paper, but we include it in the dataset release for future work. One such example is shown in Figure A2. Furthermore, the distributions of unknown labels are visualized in Figure A3.

### A.3.3 Positive and Negative Label Statistics

In Table A6 we give positive and negative label statistics across all splits of each datasets. All images in L48 contain at least 24 negative labels derived from metadata discussed in Section A.2.2. We also plot the distributions of positives, negatives, and unknowns individually for each dataset in Figure A3. L48 shows a bimodal distribution for negatives and unknowns, because each image is either fully-labeled or labels are generated through metadata. The negative labels generated by checklist and location data vary, but on average around 45 negative labels can be generated through this method.

| Known Labels | # Images | % Images |
|---|---|---|
| Fully-labeled | 38,975 | 45.75% |
| At least one box | 45,178 | 53.03% |
| Any labels | 85,193 | 100% |

Table A5: L48 degree of annotation for the training set. All images contain negative labels generated from checklist and geographic information, but positives labels must be manually labeled. Images with "Unknown Bird" labels are not considered fully-labeled.

| Dataset | Split | # Images | + (min) | + (max) | + (avg) | + (med) | - (min) | - (max) | - (avg) | - (med) |
|---------|-------|----------|---------|---------|---------|---------|---------|---------|---------|---------|
| VOC | Train | 4574 | 1 | 5 | 1.46 | 1 | 15 | 19 | 18.54 | 19 |
| VOC | Val | 1143 | 1 | 5 | 1.46 | 1 | 15 | 19 | 18.54 | 19 |
| VOC | Test | 5823 | 1 | 5 | 1.43 | 1 | 15 | 19 | 18.57 | 19 |
| VOC | All | 11540 | 1 | 5 | 1.45 | 1 | 15 | 19 | 18.55 | 19 |
| COCO | Train | 65665 | 1 | 18 | 2.94 | 2 | 62 | 79 | 77.06 | 78 |
| COCO | Val | 16416 | 1 | 16 | 2.92 | 2 | 64 | 79 | 77.08 | 78 |
| COCO | Test | 16416 | 1 | 16 | 2.92 | 2 | 64 | 79 | 77.08 | 78 |
| COCO | All | 98497 | 1 | 18 | 2.93 | 2 | 62 | 79 | 77.07 | 78 |
| L48 | Train | 85193 | 0 | 8 | 0.84 | 1 | 24 | 100 | 69.59 | 59 |
| L48 | Train+ | 45178 | 1 | 8 | 1.58 | 1 | 24 | 99 | 85.33 | 98 |
| L48 | Val | 12448 | 0 | 7 | 0.81 | 1 | 25 | 100 | 67.72 | 53 |
| L48 | Test | 31365 | 0 | 8 | 0.78 | 0 | 24 | 100 | 68.73 | 53 |
| L48 | All | 129006 | 0 | 8 | 0.82 | 1 | 24 | 100 | 68.47 | 56 |

Table A6: An overview of each datasets' positive and negative labels in terms of minimum per image, maximum per image, average, and median for training, validation, and testing splits as well as all three splits combined. "+" signifies the number of positive labels and "-" signifies the number of negative labels. The number of unknown labels can implicitly be calculated using these two values by subtracting by the total number of classes for the dataset. For L48, "Train+" signifies the training set with images with at least one positive. On VOC and COCO, the validation sets used are the ones used in our experiments, which are a randomly selected subset of the original training set.

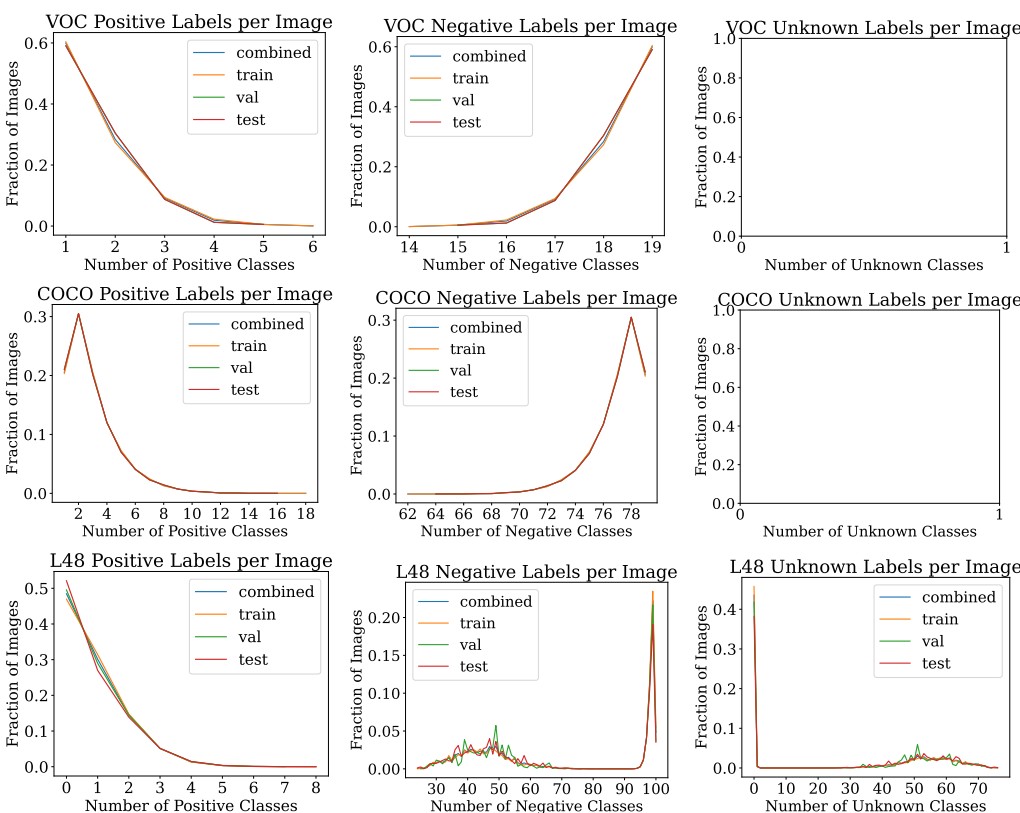

Figure A3: Visualization of dataset positives, negatives, and unknown labels per image for each split and the combined splits. For COCO and VOC, unknown label graphs are left blank because all images in these datasets are fully-labeled.

| Method | Dataset | Learning Rate | Method Hyperparameter |
|--------|---------|---------------|----------------------|
| BCE | COCO | $1e-5$ | N/A |
| BCE-AN | COCO | $1e-5$ | N/A |
| WAN | COCO | $1e-5$ | $\gamma = 1/79$ |
| LS | COCO | $1e-5$ | $\epsilon = 0.1$ |
| ROLE | COCO | $1e-5$ | $\lambda = 1$ |
| EM | COCO | $1e-5$ | $\alpha = 0.1$ |
| LL-R | COCO | $1e-5$ | $\Delta_{\text{rel}} = 0.4$ |
| LL-Ct | COCO | $1e-5$ | $\Delta_{\text{rel}} = 0.2$ |
| LL-Cp | COCO | $1e-5$ | $\Delta_{\text{rel}} = 0.2$ |
| BCE | L48 | $1e-4$ | N/A |
| BCE-AN | L48 | $1e-4$ | N/A |
| WAN | L48 | $1e-4$ | $\gamma = 1/99$ |
| LS | L48 | $1e-4$ | $\epsilon = 0.1$ |
| ROLE | L48 | $1e-4$ | $\lambda = 1$ |
| EM | L48 | $1e-4$ | $\alpha = 0.2$ |
| LL-R | L48 | $1e-4$ | $\Delta_{\text{rel}} = 0.1$ |
| LL-Ct | L48 | $1e-4$ | $\Delta_{\text{rel}} = 0.1$ |
| LL-Cp | L48 | $1e-4$ | $\Delta_{\text{rel}} = 0.1$ |

Table A7: Testing hyperparameters used for the target-only regime.

## A.4 Hyperparameters

We use mean average precision (mAP, *i.e.* the mean of per-class average precision), as our evaluation metric. For COCO, we use 20% of the training set as a validation set for hyperparameter tuning. For the L48, we select 10 training assets per species to make up the validation set which are specified in the repository. Since L48 has incomplete labels, we calculate mAP only using images which have labels for all species.

Following the training procedure of prior work [7, 18, 47], each of our experiments was trained using an ImageNet [10] pretrained ResNet50 [12] architecture using the Adam optimizer on Pytorch. Prior works [4, 37] have shown substantial improvements from ImageNet pretraining for spectrogram classification despite the domain shift. We preprocess each image by resizing the image to shape (448, 448) and normalizing the image to ImageNet statistics. For COCO only, at training time, we flip the image horizontally at random. We train for 10 epochs using a fixed batch size of 16 and a constant learning rate, which we sweep using values in $\{1e-2, 1e-3, 1e-4, 1e-5\}$. For WAN on L48 we found convergence was slower so we trained these experiments for 20 epochs instead of 10. We monitor performance on the validation set, and the best performing configuration is used for evaluation on the test set. For other SPML methods, to reduce the amount of trials required for hyperparameter tuning, we first tune the learning rate of each loss function with the hyperparameters reported for each method on COCO before sweeping the suggested range of hyperparameters given in each respective work. Once these settings are chosen, each experiment is repeated 5 times to calculate mean and standard deviation performance. The settings used in each of our experiments can be found in Tables A7, A8, and A9. For COCO experiments, we use a different randomly-generated SPML dataset each time, though these are the same across methods. For L48 experiments, we only train with images containing at least one positive, meaning we remove images with only confirmed negatives. Following [18], we increase the learning rate of the last layer by 10x for training the LL-variants.

For $\mathcal{R}_P$ hyperparameters, we run a grid search with $\alpha \in \{1e-1, 1e-2, 1e-3\}, \epsilon \in \{1e-2, 1e-3, 1e-4\}$. We initialize $\overline{y_0^i}$ following ROLE initialization [7], $\overline{y_0^i} \sim \mathcal{U}(0.4, 0.6)$.

| Method | Dataset | Learning Rate | Method Hyperparameter | $\alpha$ | $\epsilon$ |
|---|---|---|---|---|---|
| BCE | L48 | $1e-4$ | N/A | $1e-1$ | $1e-2$ |
| BCE-AN | L48 | $1e-4$ | N/A | $1e-1$ | $1e-3$ |
| WAN | L48 | $1e-4$ | $\gamma = 1/99$ | $1e-2$ | $1e-3$ |
| LS | L48 | $1e-4$ | $\epsilon = 0.1$ | $1e-2$ | $1e-3$ |
| ROLE | L48 | $1e-4$ | $\lambda = 1$ | $1e-1$ | $1e-4$ |
| EM | L48 | $1e-4$ | $\alpha = 0.2$ | $1e-1$ | $1e-4$ |
| LL-R | L48 | $1e-4$ | $\Delta_{\text{rel}} = 0.1$ | $1e-1$ | $1e-2$ |
| LL-Ct | L48 | $1e-4$ | $\Delta_{\text{rel}} = 0.1$ | $1e-2$ | $1e-2$ |
| LL-Cp | L48 | $1e-4$ | $\Delta_{\text{rel}} = 0.1$ | $1e-2$ | $1e-4$ |

Table A8: Testing hyperparameters used for asset regularization.

| Method | Dataset | Learning Rate | Method Hyperparameter | a | b |
|---|---|---|---|---|---|
| WAN | COCO Geo | $1e-5$ | $\gamma = 0.1$ | 0 | 0.01 |
| LS | COCO Geo | $1e-5$ | $\epsilon = 0.2$ | 0 | 0.05 |
| ROLE | COCO Geo | $1e-5$ | $\lambda = 0.1$ | 0 | 0.01 |
| EM | COCO Geo | $1e-5$ | $\alpha = 0.1$ | 1 | 0.01 |
| LL-R | COCO Geo | $1e-5$ | $\Delta_{\text{rel}} = 0.4$ | N/A | N/A |
| LL-Ct | COCO Geo | $1e-5$ | $\Delta_{\text{rel}} = 0.2$ | N/A | N/A |
| LL-Cp | COCO Geo | $1e-5$ | $\Delta_{\text{rel}} = 0.2$ | N/A | N/A |
| WAN | COCO Checklist | $1e-5$ | $\gamma = 0.1$ | 1 | 0.01 |
| LS | COCO Checklist | $1e-5$ | $\epsilon = 0.1$ | 1 | 0.5 |
| ROLE | COCO Checklist | $1e-5$ | $\lambda = 1$ | 1 | 1 |
| EM | COCO Checklist | $1e-5$ | $\alpha = 0.1$ | 1 | 0.02 |
| LL-R | COCO Checklist | $1e-5$ | $\Delta_{\text{rel}} = 0.4$ | N/A | N/A |
| LL-Ct | COCO Checklist | $1e-5$ | $\Delta_{\text{rel}} = 0.2$ | N/A | N/A |
| LL-Cp | COCO Checklist | $1e-5$ | $\Delta_{\text{rel}} = 0.2$ | N/A | N/A |
| WAN | L48 Geo | $1e-4$ | $\gamma = 0.05$ | 1 | 0.5 |
| LS | L48 Geo | $1e-4$ | $\epsilon = 0.1$ | 1 | 0.2 |
| ROLE | L48 Geo | $1e-4$ | $\lambda = 0.5$ | 0 | 0.05 |
| EM | L48 Geo | $1e-4$ | $\alpha = 0.1$ | 0 | 0.01 |
| LL-R | L48 Geo | $1e-4$ | $\Delta_{\text{rel}} = 0.1$ | N/A | N/A |
| LL-Ct | L48 Geo | $1e-4$ | $\Delta_{\text{rel}} = 0.1$ | N/A | N/A |
| LL-Cp | L48 Geo | $1e-4$ | $\Delta_{\text{rel}} = 0.1$ | N/A | N/A |
| WAN | L48 Checklist | $1e-4$ | $\gamma = 1/99$ | 0 | 0.5 |
| LS | L48 Checklist | $1e-4$ | $\epsilon = 0.1$ | 1 | 1 |
| ROLE | L48 Checklist | $1e-4$ | $\lambda = 2$ | 0 | 0.05 |
| EM | L48 Checklist | $1e-4$ | $\alpha = 0.02$ | 1 | 0.01 |
| LL-R | L48 Checklist | $1e-4$ | $\Delta_{\text{rel}} = 0.1$ | N/A | N/A |
| LL-Ct | L48 Checklist | $1e-4$ | $\Delta_{\text{rel}} = 0.1$ | N/A | N/A |
| LL-Cp | L48 Checklist | $1e-4$ | $\Delta_{\text{rel}} = 0.1$ | N/A | N/A |

Table A9: Testing hyperparameters used for the geo/checklist regime.

| Method | $\mathcal{L}^+$ | $\mathcal{L}^?$ |
|---|---|---|
| BCE | $-\log(f_\theta^i)$ | $-\log(1-f_\theta^i)$ |
| BCE-AN | $\mathcal{L}_{\text{BCE}}^+$ | $\mathcal{L}_{\text{BCE}}^-$ |
| WAN | $\mathcal{L}_{\text{BCE}}^+$ | $\gamma\mathcal{L}_{\text{BCE}}^-$ |
| LS | $\frac{1-\epsilon}{2}\mathcal{L}_{\text{BCE}}^+ + \frac{\epsilon}{2}\mathcal{L}_{\text{BCE}}^-$ | $\frac{1-\epsilon}{2}\mathcal{L}_{\text{BCE}}^- + \frac{\epsilon}{2}\mathcal{L}_{\text{BCE}}^+$ |
| ROLE | See [7] | See [7] |
| EM | $\mathcal{L}_{\text{BCE}}^+$ | $-\alpha(f_\theta^i\mathcal{L}_{\text{BCE}}^+ + (1-f_\theta^i)\mathcal{L}_{\text{BCE}}^-)$ |
| LL-R | $\mathcal{L}_{\text{BCE}}^+$ | $\mathbb{1}_{[\neg\text{LL}]}\mathcal{L}_{\text{BCE}}^-$ |
| LL-Ct | $\mathcal{L}_{\text{BCE}}^+$ | $\mathbb{1}_{[\neg\text{LL}]}\mathcal{L}_{\text{BCE}}^- + \mathbb{1}_{[\text{LL}]}\mathcal{L}_{\text{BCE}}^+$ |
| LL-Cp | $\mathcal{L}_{\text{BCE}}^+$ | $\mathbb{1}_{[\neg\text{LL}]}\mathcal{L}_{\text{BCE}}^- + \mathbb{1}_{[\text{LL}]}\mathcal{L}_{\text{BCE}}^+$ |

Table A10: Positive and unknown losses for the SPML methods. For BCE row, $\mathcal{L}^?$ signifies $\mathcal{L}^-$ since BCE is trained on full labels. The variables $\gamma, \epsilon, \alpha$ are all hyperparameters for each respective method. For the LL-variants, LL signifies whether the loss term falls in the top $((t-1)\cdot\Delta)\%$ of losses in the batch.

| Method | L48 | L48 $+\mathcal{R}_P$ | L48 $+\mathcal{R}_E$ |
|---|---|---|---|
| BCE | $62.44 \pm 0.51$ | (+3.93) $66.37 \pm 0.48$ | (+0.59) $63.03 \pm 0.42$ |
| BCE-AN | $52.23 \pm 0.48$ | (+3.83) $56.06 \pm 1.11$ | (+0.12) $52.35 \pm 0.54$ |
| WAN | $51.96 \pm 0.55$ | (+3.70) $55.66 \pm 0.72$ | (-0.07) $51.89 \pm 0.45$ |
| LS | $\mathbf{56.42 \pm 0.67}$ | (+0.02) $\mathbf{56.44 \pm 0.73}$ | (-0.08) $\mathbf{56.34 \pm 0.51}$ |
| ROLE | $54.00 \pm 0.95$ | (+0.14) $54.14 \pm 0.54$ | (-0.51) $53.49 \pm 0.73$ |
| EM | $55.27 \pm 0.97$ | (-0.08) $55.19 \pm 0.66$ | (+0.35) $55.62 \pm 0.21$ |
| LL-R | $50.06 \pm 0.79$ | (+4.90) $54.96 \pm 0.35$ | (+0.07) $50.13 \pm 0.84$ |
| LL-Ct | $47.98 \pm 0.90$ | (+6.11) $54.09 \pm 0.55$ | (+2.09) $50.07 \pm 1.41$ |
| LL-Cp | $43.80 \pm 0.80$ | (+0.60) $44.40 \pm 0.58$ | (+0.58) $44.38 \pm 0.61$ |

Table A11: Compiled mAP results (given in percentages) on the test set for each method, averaged across five runs for unmodified, probability regularized, and embedding regularized SPML methods.

## A.5 Additional Experiments and Analysis

### A.5.1 Embedding Asset Regularization

We extend the idea of asset-level similarity to the embedding level, by enforcing embeddings of a clip to be similar to the average embeddings across the entire asset, with a regularization term $\mathcal{L}_E$:

$$\mathcal{R}_E(x_j^i) = \text{MSE}(d_j^i, \overline{d_t^i}) \tag{1}$$

where MSE is mean-squared error and $d_j^i$ is the last layer embedding of the network for the $j$-th clip of recording $i$. We use a similar equation to calculate the running average of the embedding as for the probability asset regularization but use a different $\epsilon_2$.

**Probability regularization is generally more effective than embedding regularization.** In Table A11, we see the average performance boost for embedding regularization is much less than the boost for probability regularization. We attribute this to the recurrence of background species at the asset-level giving an accurate and more direct training signal. The supervision provided at the prediction-level is a strong prior because species positives in one clip are very likely to reoccur. In contrast, embedding regularization is more indirect than probability regularization, as the model can learn spurious correlations at the embedding level like fixed background noise within an asset. Regardless, we do find that embedding regularization still has a minor positive effect on training, potentially working as a weaker form of $\ell_2$ weight decay to prevent overfitting on noisy target-only data.

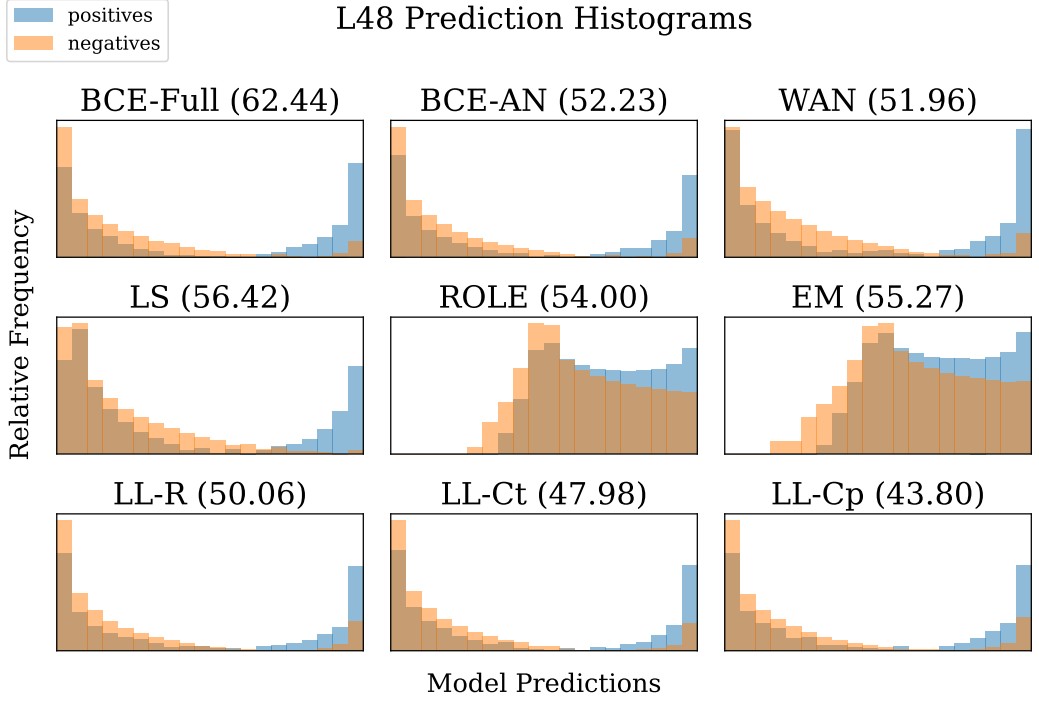

Figure A4: Histogram of model outputs, log-scaled, shown separately for positive and negative labels on the test set of L48. We see ROLE and EM have significantly different distributions from the other methods and the LL-variants all have higher rates of high confidence false negatives compared to the other methods.

### A.5.2 Model Output Histograms

In Figure A4, we show the model prediction distributions for positive and negative labels on the L48 test set. We see a higher rate of high confidence false positives for the LL methods and a significantly shifted probability distribution for ROLE and EM.

### A.5.3 Analysis of Specific Species

In Figure A5, we include PR curves for the LL-R method in the three data regimes and with regularization for five different species. Interestingly, we see different patterns for the two groups of species. In the three plots on the left, we compare the PR curves for Carolina Chickadee, Black-capped Chickadee, and Mountain Chickadee, which all are geographically separated but are vocally similar. We see that providing negative labels through geographical priors gives a large increase to the model's precision, indicating the labels are helping with this fine-grained confusion. In contrast, in the two right plots we see the opposite effect. Yellow-bellied Sapsucker and Red-breasted Sapsucker are also geographically separated and nearly vocally identical, but we see providing the model with negative labels through geographical priors decreases performance. Our hypothesis for this distinction is the model is unable to learn the sapsucker task because it is more difficult than the chickadee task. In the chickadee task, the species are similar-sounding, but have known differences in vocal patterns. As a result, providing the model explicit negatives prevents LL-R from rejecting the losses for the similar chickadees and the model learns to distinguish the two. In contrast, the sapsucker task is more difficult than the chickadee task, as the two species do not have distinctive vocal differences. As a result, rejecting the loss in this case may prevent the model from being forced to learn an intractable task and instead allow it to accept ambiguity on these two species instead of having to predict confidently.

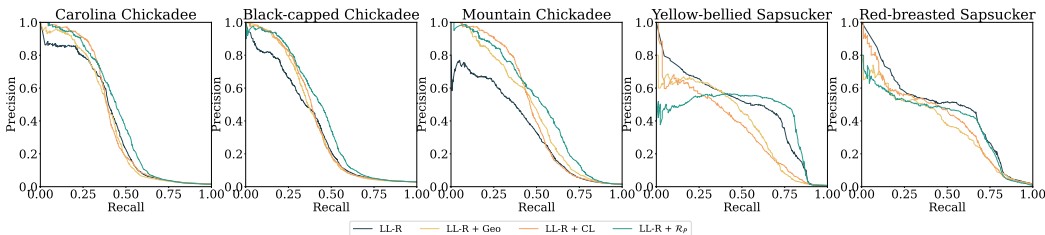

Figure A5: Precision-recall curves for the LL-R method in target-only, target-only with regularization, geo, and checklist regimes for five different species, where the species name is given as the graph title.

