# OpenReview forum: "Merlin L48 Spectrogram Dataset"
_NeurIPS.cc/2025/Datasets_and_Benchmarks_Track — NeurIPS 2025 Datasets and Benchmarks Track poster_

### Official Review · Reviewer_3wLh · 2025-06-02

**Rating:** 4
**Confidence:** 4

**Summary:**

The paper introduces the Merlin Lower 48 Spectrogram (ML48S) Dataset, a comprehensive collection designed to address the challenges in Single-Positive Multi-Label (SPML) learning within a real-world, fine-grained context. Unlike existing SPML benchmarks that rely on synthetic modifications of fully-annotated datasets, ML48S offers naturally occurring single-positive labels derived from extensive bird sound recordings across the contiguous United States. ML48S serves as a robust benchmark for evaluating SPML methods in realistic settings, highlighting the limitations of current approaches and guiding future research towards more adaptable and context-aware algorithms. The dataset also opens avenues for exploring semi-supervised learning and active learning strategies to better utilize partial and weak labels.

**Dataset Code Accessibility:**

Yes

**Dataset Code Comments:**

The dataset is collected from the contiguous United States (the “Lower 48”). Thus it is believed that it can be accessed.

**Ethical Considerations:**

No, there are no or only very minor ethics concerns

**Final Justification:**

I have no further comments at this point.

**Limitations Weaknesses:**

1.Despite introducing novel regularization techniques, SPML methods evaluated on ML48S still exhibit a significant performance gap compared to fully supervised approaches. This indicates that current SPML methodologies may not yet effectively bridge the gap between partial and complete label information in complex, real-world settings. Moreover, the performance boost of the proposed loss is slightly marginal.

2.The consistency-based regularization scheme introduced is designed to exploit the temporal structure inherent in spectrogram recordings. While effective for ML48S, this approach may not generalize to datasets lacking similar temporal or structural characteristics.

3.The unique challenges presented by bioacoustic data, such as overlapping vocalizations and temporal consistency, differ from those in typical image-based multi-label tasks. Consequently, the insights and methods derived from ML48S may not directly translate to other multi-label classification problems without significant adaptation.

4.ML48S is tailored specifically for bird sound classification within a particular ecological context. This specialization may limit its utility for SPML research in other domains, such as medical imaging or general object recognition, where label structures and challenges differ significantly.

5.The benchmarking primarily evaluates a set of existing SPML methods, some of which underperform on ML48S. However, the exploration of newer or alternative methods tailored to fine-grained and ecologically complex datasets is limited.

6.With data from 100 bird species, the dataset, while substantial, does not capture the full biodiversity present in larger or more ecologically diverse regions. This limitation may affect the applicability of SPML methods developed using ML48S to broader or more specialized taxonomic groups.

7.The ML48S dataset encompasses recordings from only the contiguous 48 states of the United States. This narrow geographical scope may hinder the generalizability of the findings to other regions with different ecological dynamics and bird species.

**Strengths Contributions:**

1. The proposed dataset ML48S fills the gap between large-scale archive datasets and smaller, specialized collections by providing dense, fine-grained labels suitable for SPML and detection-based tasks.

2. The authors evaluate existing SPML algorithms on ML48S, revealing significant performance discrepancies compared to synthetic datasets. Methods that performed well on benchmarks like COCO struggled with the fine-grained complexities of ML48S.

3.To mitigate misclassifications inherent in real-world data, the paper proposes a novel regularization technique that enforces temporal consistency across recordings. This approach enhances the performance of various SPML methods on the ML48S dataset.

---

> ### Author Rebuttal · Authors · 2025-07-29
>
> We thank reviewer 3wLh for their detailed review. We appreciate the recognition of our dataset’s contribution and analysis of existing methods. We want to address and clarify some of the weaknesses the reviewer brings up:
>
> **Limited geographical and species coverage (also addressed to reviewer ZAEy)**\
> We agree a broader scope is crucial for understanding global environmental trends and habitats. Our focus for ML48S was to create the first densely-annotated, year-round dataset spanning an entire country. Unfortunately, collecting dense labels for a broader range is very difficult. Datasets such as iNatSounds and BirdSet which do not contain dense labels suffer from lack of recordings in areas outside of North American and Western Europe, and the number of experts in these regions is much more limited. By concentrating on the Lower 48, we ensured higher-quality annotations and recordings which can capture real-world challenges such as regional and seasonal variations in a manageable and diverse benchmark.
>
> **Performance gap and regularization efficacy/generalizability**\
> The reviewer correctly identifies the gap between SPML methods and the fully-supervised model performance. We argue this is one of our work’s key findings rather than a limitation. ML48S reveals SPML on a real-world, fine-grained task is much harder than synthetic benchmarks suggest, and existing methods are insufficient for these settings.
>
> Asset regularization is a first step to addressing this. While we acknowledge the improvement for the best method (LS) is marginal, we want to highlight that regularization boosts the baseline BCE-AN from 52.2% to 56.1%, outperforming nearly all other more complex SPML methods. This demonstrates that leveraging domain structure is a crucial and promising strategy for tackling realistic and challenging tasks.
>
> Finally, while asset regularization is designed for the temporal structure of audio, the use of structural relationships in training is broadly applicable across domains. Because of the strict input requirements for image-based neural networks, many tasks with other data types contain inter-image relationships. For instance, satellite images are related based on their proximity to one another similar to different clips in a longer recording, or a medical scan might be broken up into different images due to being too large to input into a single network. As a result, ML48S is an excellent realistic benchmark for testing consistency-based methods.
>
> **New and alternative methods for fine-grained ecological data**\
> Our primary goal was to benchmark the core SPML field, which has been predominantly evaluated on coarse-grained datasets like COCO. ML48S was well-suited to test these SPML methods in a realistic, fine-grained setting where they have not been tested before. We agree methods like TaxaBind [1] are promising future directions and expect their pretrained model to have strong zero-shot performance for our task, but we found ML48S to be most insightful in revealing the limitations of the established SPML literature.
>
> [1] Srikumar Sastry, Subash Khanal, Aayush Dhakal, Adeel Ahmad, and Nathan Jacobs. Taxabind: A unified embedding space for ecological applications. In Proceedings of the IEEE/CVF Winter Conference on Applications of Computer Vision, 2025.

---

> > ### Author Response · Authors · 2025-08-05
> >
> > We again thank Reviewer 3wLh again for their detailed and comprehensive review. In our rebuttal, we aimed to thoroughly address the important points you raised regarding performance gaps, the generalizability of our methods, and the scope of the work. We hope our clarifications are helpful for your final evaluation and would be happy to engage in any further discussion. Thank you for your valuable time and consideration.

---

> > > ### Comment · Reviewer_3wLh · 2025-08-08
> > >
> > > Thank you for the detailed responses and clarifications. I have no further comments at this point. I will raise my score.

---

### Official Review · Reviewer_fGqE · 2025-06-29

**Rating:** 6
**Confidence:** 4

**Summary:**

This paper introduces a new real-world dataset, ML48S, designed to advance state-of-the-art methods for single-positive multi-label (SPML) learning. The dataset is exhaustively annotated with bounding boxes on audio spectrograms of bird sounds, encompassing 100 species and covering 100 hours of soundscapes across the lower 48 states of the US throughout the year. ML48S is the first dataset of its kind, presenting real-world challenges not captured by previous synthetic SPML datasets derived from sources like COCO. The authors demonstrate that methods typically effective on synthetic datasets with distinct classes often fail on ML48S due to its fine-grained nature, featuring many similar species that sound alike but are distinct.

Additionally, the authors conduct an ablation study on various state-of-the-art SPML methods. They show that incorporating consistency-based regularization during training improves prediction consistency within a recording, thereby enhancing the performance of all prior SPML loss functions on ML48S. The code and dataset are publicly available.

**Additional Feedback:**

Regarding lines 287-290, I believe there may be a mistake. It appears that in Figure 5-d the recall is decreasing rather than increasing at high precision for LL-Ct when the regularization is applied. Can you recheck this analysis?

**Dataset Code Accessibility:**

Yes

**Dataset Code Comments:**

The authors have provided clear code and a link for downloading the data.

**Ethical Considerations:**

No, there are no or only very minor ethics concerns

**Final Justification:**

After reading the rebuttal, I believe the authors did excellent work. I'm keeping my score to 6: strong accept.

**Limitations Weaknesses:**

The work is excellent and has no major weaknesses. However, I have a minor comment:

The consistency-based regularization method assumes that background species within an asset tend to repeat across multiple images. While this is a reasonable assumption, it may not hold true for all scenarios, particularly in dynamic environments where bird sounds are transient.

**Strengths Contributions:**

The introduction of the ML48S dataset is a significant contribution as it addresses a gap in the availability of real-world datasets for SPML tasks. Unlike synthetic datasets, ML48S captures the complexities and nuances of real-world scenarios, particularly in the domain of bird species identification.

The paper benchmarks existing SPML methods on the ML48S dataset and compares their performance to synthetic SPML datasets derived from COCO. This comparison highlights the limitations of current methods when applied to real-world, fine-grained datasets.

The introduction of consistency-based regularization is a novel approach to improving the performance of SPML methods. This technique promotes inter-recording similarity, leveraging the temporal consistency inherent in audio recordings.

The release of clear, well-documented code and the dataset itself is a significant contribution to the reproducibility and transparency of the research. This allows other researchers to build upon the work.

---

> ### Author Rebuttal · Authors · 2025-07-29
>
> We thank reviewer fGqE their positive feedback and support of our work. We are grateful for their careful reading and have the following responses to their comments:
>
> **Recurrence of background species**\
> The reviewer brings up an astute observation, that background species are not necessarily recurring across different clips within an asset. While not universally true, we found this to be empirically well-supported in ML48S. Namely, a background species which is present in an asset appears, on average, in 28% of the clips from that same asset. This high recurrence rate provides a reliable signal for asset regularization.
>
> **Figure 5d and lines 287-290**\
> We thank the reviewer for their careful reading and giving us a chance to re-check our analysis. There may have been a misunderstanding of our claims regarding Figure 5d. When we compare the solid gray line (unregularized LL-Ct) with the dotted gray line (regularized LL-Ct) at high precisions (around 0.8-1.0), we notice a marked increase in recall. For instance, at precision 0.9, the unregularized method shows a recall just above 0.05, while the regularized method has a recall of about 0.2. We hope this clarification is helpful.

---

> > ### Comment · Reviewer_fGqE · 2025-08-05
> >
> > I thank the authors for their rebuttal which addresses my minor comments. Excellent work!

---

### Official Review · Reviewer_ZAEy · 2025-06-30

**Rating:** 4
**Confidence:** 3

**Summary:**

This paper provides a realistic benchmark for single-positive multi-label (SPML) learning using bird audio spectrograms. Its fine-grained labels, ecological relevance, and proposed asset regularization method make it valuable for SPML research. However, broader geographic/species coverage and deeper analysis of method failures would strengthen the work. Recommended for acceptance, with suggestions for additional failure case studies.

**Dataset Code Accessibility:**

Yes

**Ethical Considerations:**

No, there are no or only very minor ethics concerns

**Limitations Weaknesses:**

The dataset’s geographic scope (U.S.-only) and species coverage could be expanded to improve ecological relevance. Some SPML methods (e.g., LL variants) perform poorly on ML48S, suggesting a need for deeper failure analysis or adaptive loss functions. The paper would benefit from exploring how to better leverage negative labels beyond simple exclusion.

**Strengths Contributions:**

This work presents a realistic single-positive multi-label (SPML) benchmark derived from bird audio recordings, addressing a key gap in synthetic SPML datasets. The dataset’s fine-grained labels (100 species across seasons/regions) and dense annotations (including background species) enable rigorous evaluation of SPML methods. The proposed asset regularization, which enforces temporal consistency within recordings, is simple yet effective. The inclusion of geographic and checklist-derived negative labels further enriches the learning framework.

---

> ### Author Rebuttal · Authors · 2025-07-29
>
> We thank reviewer ZAEy for their time and efforts in reviewing our work. We are pleased to see the reviewer found our dataset valuable in addressing weaknesses with synthetic SPML datasets and our exploration of asset regularization and negative labels to be productive. We want to address some of the limitations brought up by the reviewer below:
>
>
> **Geographical and species scope (also addressed to reviewer 3wLh)**\
> We agree that a broader geographical and taxonomic scope is crucial for understanding global environmental trends and habitats. Our focus for ML48S was to create the first densely-annotated, year-round dataset spanning an entire country. Unfortunately, collecting dense labels for a broader scale  is very difficult. Datasets such as iNatSounds and BirdSet which do not contain dense labels suffer from lack of recordings in areas outside of North American and Western Europe, and the number of experts in these regions is equally limited. By concentrating on the Lower 48, we ensured higher-quality annotations and recordings which can capture real-world challenges such as regional and seasonal variations in a manageable and diverse benchmark.
>
> **Failure analysis of LL-variants**\
> We share the reviewer’s surprise at the poor performance of LL-variants. Our analysis in Sections 5.2 and 5.3 points to an incorrect assumption likely being made as a result of fine-grained confusions in the ML48S. We believe LL-variants attribute these confusions to mislabeled negatives, while in ML48S high model confidence on unknown labels arise from model misclassifications between two similar-sounding species. Appendix A.5.3 adds nuance to this discussion, examining two sets of similar species (chickadees and sapsuckers) which have opposite interactions with negative labels on LL-methods.
>
> **Better leveraging of negative labels**\
> The reviewer's observation is excellent and agrees with our finding that current SPML methods are not well-equipped to handle mixed-supervision settings. As in Section 5.4, we found incorporating known negatives by modifying existing SPML losses led to inconsistent gains even with extensive hyperparameter tuning, indicating a lack of flexibility. We believe that this limitation has gone largely unnoticed in prior work due to the use of synthetic benchmarks which lack negative labels obtained from domain priors. ML48S reveals this new setting as a promising direction for future research.

---

> ### Comment · Reviewer_ZAEy · 2025-08-09
>
> Thank you for your response. I believe this paper makes a significant contribution by presenting a novel and promising research direction with practical applications. Therefore, I support the acceptance of this paper.

---

### Official Review · Reviewer_67YZ · 2025-07-01

**Rating:** 4
**Confidence:** 2

**Summary:**

This paper proposes a partially labeled dataset. Specifically, the dataset is labeled with single-positive label while there might be other labels missing (or Single-Positive Multi-label, SPML). The dataset is captured and manually labeled and the authors demonstrated that the proposed dataset can capture more natural distribution shift while existing synthetic dataset cannot. This lead to a more realistic study on the SPML setting and the corresponding algorithms.

**Dataset Code Accessibility:**

Yes

**Dataset Code Comments:**

The dataset and the codebase are accessible to me. The codebase is well-documented.

**Ethical Considerations:**

No, there are no or only very minor ethics concerns

**Final Justification:**

Some of my concerns (e.g., about the experimental and data collection setting) are addressed.

**Limitations Weaknesses:**

I'm mostly confused about the SPML setting. In the synthetic dataset, the behavior model of simulated annotators are clear, e.g., they drop labels based on some stochastic distribution. Although unrealistic, it is intuitively easy to understand how labels are missing. In the proposed dataset, the situation is kind of arbitrary -- we don't know what the annotators did in the annotation procedure. If another group of people would like to take some effort and reproduce the dataset collection procedure, but with a different group of annotators, and perhaps on a different set of data samples, what can guarantee a similar resulting dataset? In another word, I believe the resulting dataset is **very** subjective to the specific group of annotators, or even a specific amount of payment per data sample.

**Strengths Contributions:**

- The motivation of constructing this dataset is well explained.

- The dataset is aiming to solve a realistic issue for reducing data annotation cost.

- Thorough experiments on how existing baselines work on the proposed dataset is presented.

---

> ### Author Rebuttal · Authors · 2025-07-29
>
> We thank reviewer 67YZ for their time and effort in reviewing our work. We are happy to hear the motivations of the dataset were clear and experimentation and benchmarking are thorough.
>
> We would like  to clarify any ambiguity in our target-only SPML setting regarding its construction, annotation, and replicability. The ML48S dataset construction is a systematic process that results in a naturally-occurring SPML task, which we argue is a key advantage over synthetic dataset. The ML48S is generated from the following process:
>
> 1. **Recording selection via target species**: Raw audio recordings are collected from eBird, meaning initially they only contain metadata such as location and date and a single “target species” label. For each species, we selected 100 recordings based on the target species label, which guarantees a positive label for this species. Since these recordings may also contain additional unlabeled background species, this is a natural SPML scenario from the outset. Datasets like BirdSet and iNatSounds stop at this level of annotation.
> 2. **Dense annotation**: A small set of experts perform dense annotations, identifying all vocalizing species with bounding boxes. For longer recordings, annotators fully-label segments of the recording instead of the entire recording, which tends to be redundant.
> 3. **Target-only training**: Our "target-only" training regime directly uses the initial "target species" as the single positive label, and discards all background species. This setup precisely simulates a real-world workflow where annotators can quickly and reliably identify one species they are confident in while ignoring background species.
>
> We believe this process is objective and replicable in domains where sparse labels are easier to obtain than exhaustive ones. For example, the target-only dataset used in our SPML task can be created directly by having annotators label only the target species in step 2. This greatly expands the pool of viable annotators.
>
> For bird sounds, we found annotation expertise is often geographically localized—an expert may recognize bird sounds from their region but may struggle with unfamiliar sounds from distant areas.  Requiring full annotations across species can thus reduce annotation throughput. In contrast, if they only annotate species they are confident about, they can quickly annotate recordings even from outside their region of expertise. Hence, by only annotating the target species, the dataset can be annotated more quickly and accurately by a larger set of annotators.
>
> We hope this clarifies how our dataset can be constructed in a consistent and objective procedure, as well as explaining why and how groups might create target-only SPML data for their specific needs.

---

> > ### Author Response · Authors · 2025-08-05
> >
> > We thank reviewer 67YZ again for their thoughtful review. In our rebuttal, we aimed to clarify how the dataset construction process is systematic and objective. We would be very grateful for your feedback on whether our clarification was helpful or if you have any further questions. Thank you for your time and consideration.

---

> > ### Comment · Reviewer_67YZ · 2025-08-08
> > **feedback**
> >
> > Thanks for the authors feedback. It did address some of my concerns.
> >
> > The remaining questions / concerns are:
> > - As you mentioned "annotation expertise is often geographically localized", I wonder if there are any statistics about the geographic distribution of all annotators. Do you consider the effect / bias caused by this distribution? If another research group aims to reproduce the results, but hired another group of annotators, will the conclusion / methods developed be different?
> >
> > Since some of my concerns are addressed, I'll raise my rating to 4.

---

> > > ### Author Response · Authors · 2025-08-08
> > >
> > > Thank you for your positive feedback on our rebuttal and for raising your score. We sincerely appreciate it and are happy to clarify these final points.
> > >
> > > Regarding potential annotator bias, we want to clarify our motivations. We used the general principle of "geographically localized expertise" to explain why the target-only SPML setting is so relevant for real-world applications. While no human annotation process is entirely free of subtle biases, our process was designed to mitigate this concern. Our annotators are contractors hired through the Cornell Lab of Ornithology who have high proficiency with all 100 species in the ML48S dataset, meaning regional performance variations are minimized in our ground-truth labels.
> > >
> > > Regarding reproducibility, we believe another research group would arrive at consistent findings because our dataset creation process is designed to be independent of the annotator group. In our dataset creation, target species and recordings are selected using eBird metadata before any human annotation begins. Therefore, if another group used different but similarly qualified experts, they would produce the same data. While minor variations in bounding box placement and recording selection are possible, the fundamental ground truth of species presence and absence would not change. In other words, annotators are selected based on the data, not the other way around.
> > >
> > > We hope this addresses your remaining concerns, and we thank you again for the constructive dialogue.

---

### Decision · Program_Chairs · 2025-09-18

**Decision:**

Accept (poster)

**Comment:**

This paper introduces the Merlin L48 dataset, consisting of over 10,000 recordings containing bird vocalizations of 100 species from the continental United States. These spectrograms have been fully annotated using bounding boxes, which allows the dataset to be used in single-positive multi-label (SPML) experiments. The authors evaluate a variety of SPML methods using this dataset, comparing the results to those on a synthetic SPML dataset derived from COCO. Lastly, the authors introduce a temporal consistency loss; enforcing the prior that birds are likely to vocalize repeatedly, so predictions derived from 3 s windows are likely similar across time for the same recording. The results show that many SPML methods struggle with the real-world difficulty of the ML48 dataset, the temporal consistency loss (asset regularization) helps; and deriving some negative labels from metadata (e.g., geographical priors) also helps.

All reviewers agree that this dataset is valuable in that it presents a real-world example of a SPML problem that is both challenging and realistic. The takeaway that many SPML methods fail to transfer to a realistic problem is important to the field and a signal that progress might have been artificial. Similarly, the insight that deducing some negative labels from priors can be helpful has a real impact on practitioners working with weakly labeled data. The authors made an effort to make this dataset easily accessible to the computer vision community. The authors spoke positively about the writing and thorough sets of experiments.

While the limited geographical coverage is a valid concern, I concur with the reviewers that this is an acceptable trade-off for the high density of annotations at this scale. There are concerns about how insights from this paper will translate to computer vision (e.g., the confusion and failure modes are possibly different for spectrograms and natural images) but the low performance of SPML methods developed using COCO is in itself a useful data point. I encourage the authors incorporate a discussion in the final text about how asset regularization is expected to generalize to non-audio settings.

Overall the reviewers unanimously support acceptance though and I see no strong reasons to disagree. I recommend this paper for acceptance.